# Stimulator of Interferon Genes Protein (STING) Expression in Cancer Cells: A Tissue Microarray Study Evaluating More than 18,000 Tumors from 139 Different Tumor Entities

**DOI:** 10.3390/cancers16132425

**Published:** 2024-06-30

**Authors:** Anne Menz, Julia Zerneke, Florian Viehweger, Seyma Büyücek, David Dum, Ria Schlichter, Andrea Hinsch, Ahmed Abdulwahab Bawahab, Christoph Fraune, Christian Bernreuther, Martina Kluth, Claudia Hube-Magg, Katharina Möller, Florian Lutz, Viktor Reiswich, Andreas M. Luebke, Patrick Lebok, Sören A. Weidemann, Guido Sauter, Maximilian Lennartz, Frank Jacobsen, Till S. Clauditz, Andreas H. Marx, Ronald Simon, Stefan Steurer, Eike Burandt, Natalia Gorbokon, Sarah Minner, Till Krech

**Affiliations:** 1Institute of Pathology, University Medical Center Hamburg-Eppendorf, 20246 Hamburg, Germany; a.menz@uke.de (A.M.); julia898@gmx.net (J.Z.); f.viehweger@uke.de (F.V.); s.bueyuecek@uke.de (S.B.); d.dum@uke.de (D.D.); r.schlichter@uke.de (R.S.); a.hinsch@uke.de (A.H.); c.fraune@uke.de (C.F.); c.bernreuther@uke.de (C.B.); m.kluth@uke.de (M.K.); c.hube@uke.de (C.H.-M.); ka.moeller@uke.de (K.M.); f.lutz@uke.de (F.L.); v.reiswich@uke.de (V.R.); luebke@uke.de (A.M.L.); p.lebok@uke.de (P.L.); s.weidemann@uke.de (S.A.W.); g.sauter@uke.de (G.S.); m.lennartz@uke.de (M.L.); f.jacobsen@uke.de (F.J.); t.clauditz@uke.de (T.S.C.); andreas.marx@klinikum-fuerth.de (A.H.M.); s.steurer@uke.de (S.S.); e.burandt@uke.de (E.B.); n.gorbokon@uke.de (N.G.); s.minner@uke.de (S.M.); t.krech@uke.de (T.K.); 2Pathology Department, Faculty of Medicine, University of Jeddah, Jeddah 23218, Saudi Arabia; bawahab2002@hotmail.com; 3Institute of Pathology, Clinical Center Osnabrueck, 49078 Osnabrueck, Germany; 4Department of Pathology, Academic Hospital Fuerth, 90766 Fuerth, Germany

**Keywords:** STING, immunohistochemistry, tissue microarray, human carcinoma, PD-L1

## Abstract

**Simple Summary:**

STING is a key element in the cGAS/STING cytosolic sensing pathway and several STING agonists are currently being evaluated as anticancer drugs in the field of cancer immunotherapy. This study provides a unique catalog of STING expressions in tumor cells as well as its clinical relevance and association with PD-L1 expression in tumor and inflammatory cells in more than 130 different tumor entities.

**Abstract:**

Stimulator of interferon genes protein (STING) activates the immune response in inflammatory cells. STING expression in cancer cells is less well characterized, but STING agonists are currently being evaluated as anticancer drugs. A tissue microarray containing 18,001 samples from 139 different tumor types was analyzed for STING by immunohistochemistry. STING-positive tumor cells were found in 130 (93.5%) of 139 tumor entities. The highest STING positivity rates occurred in squamous cell carcinomas (up to 96%); malignant mesothelioma (88.5%–95.7%); adenocarcinoma of the pancreas (94.9%), lung (90.3%), cervix (90.0%), colorectum (75.2%), and gallbladder (68.8%); and serous high-grade ovarian cancer (86.0%). High STING expression was linked to adverse phenotypes in breast cancer, clear cell renal cell carcinoma, colorectal adenocarcinoma, hepatocellular carcinoma, and papillary carcinoma of the thyroid (*p* < 0.05). In pTa urothelial carcinomas, STING expression was associated with low-grade carcinoma (*p* = 0.0002). Across all tumors, STING expression paralleled PD-L1 positivity of tumor and inflammatory cells (*p* < 0.0001 each) but was unrelated to the density of CD8+ lymphocytes. STING expression is variable across tumor types and may be related to aggressive tumor phenotype and PD-L1 positivity. The lack of relationship with tumor-infiltrating CD8+ lymphocytes argues against a significant IFN production by STING positive tumor cells.

## 1. Introduction

The recognition of pathogens such as viruses, bacteria, and fungi occurs through the perception of pathogen-associated molecular patterns (PAMPs) or damage-associated molecular patterns (DAMPs) through a set of specialized receptors. Exogenous DNA derived from pathogens or self-DNA in the cytosol represent highly efficient PAMPs/DAMPs, which can induce strong innate immune responses through the stimulation of downstream signaling cascades including the production of proinflammatory mediators and type I interferons (IFNs). Stimulator of interferon genes (STING) is a key component in this pathogen response system (summarized in [1]). STING “activation” is triggered by cytosolic nucleic acids derived from DNA viruses and bacteria as well as damaged self-DNA in the cyclic GMP-AMP (cGAS)/STING cytosolic DNA-sensing pathway (summarized in [2]). Activated STING acts as an adapter protein to enable, for example, phosphorylation of interferon regulatory factor 3 (IRF3) by serine/threonine protein kinase (TBK1), which leads to type I IFN activation (summarized in [1]). Due to the capability of STING to stimulate immune response, numerous STING agonists are currently being evaluated for their suitability as anticancer drugs in the field of cancer immunotherapy (summarized in [3,4]), and recent evidence indicates that the combination of STING agonists with STAT3 inhibitors can further enhance antitumor immunity [5].

Most studies evaluating the function of STING have focused on inflammatory cells, but STING can also be expressed in cancer cells, in which the role of STING is less well characterized (summarized in [6]). The available data suggest a high complexity of the STING pathway in cancer cells. For example, in vitro STING knockout and STING agonist experiments have revealed reduced, unchanged, or increased cell proliferation and survival in studies employing cell lines from different cancer types (summarized in [6]). Several recent studies have suggested that the STING pathway in cancer cells can be important for regulating the antitumor immune response. For example, programmed cell death 1 ligand 1 (PD-L1) expression in tumor cells could be stimulated by STING agonist in a cancer murine model [7], by elevated STING expression in DNA damage-response-deficient breast cancer cells [8] and by radiation-therapy-induced STING expression in human and mouse liver cancer cells [9]. Furthermore, activation of the cGAS/STING pathway led to natural killer cell (NK) infiltration [10,11], whereas an absence of STING in cancer cells was found to be associated with low numbers of NKs and CD8-positive lymphocytes [12,13]. While these findings could be explained by IFN production in STING positive tumor cells, other studies have questioned the STING-induced secretion of biologically relevant quantities of IFN by tumor cells (summarized in [6]).

Most available data on STING expression in cancer are RNA-based and, therefore, cannot distinguish the specific role of STING expression in cancer cells. Studies evaluating STING expression of tumor cells by immunohistochemistry (IHC) are so far limited in number and the obtained results suggest a rather complex and variable role of STING expression in cancer. For example, STING downregulation as compared to adjacent normal tissue has been reported in colorectal [14] and gastric carcinomas [15], while STING upregulation, as compared to adjacent normal tissue, was described in squamous cell carcinomas (SQCC) of the tongue [16]. Associations of high STING levels with unfavorable tumor features were found in carcinomas of the cervix uteri [17], ovary [18], stomach [19], colon [20], lung [21], larynx [22] tongue [16], and kidney [23], while low STING levels were related to unfavorable tumor features in carcinomas of the breast [24], head and neck [25], the urinary tract [26], lung [27,28], and stomach [15], as well as in T-cell lymphoma [29].

Considering the potential clinical and therapeutic relevance of STING expression in cancer, we aimed at a comprehensive characterization of tumor cell STING expression in human neoplasms. For this purpose, a pre-existing set of tissue microarrays (TMAs) containing more than 18,000 tumor samples from 139 different tumor types and subtypes as well as 76 non-neoplastic tissue categories were analyzed for STING expression by IHC.

## 2. Materials and Methods

### 2.1. Tissue Microarrays (TMAs)

Two sets of TMAs made from formalin-fixed, paraffin-embedded tissue samples were employed in our study. These included a normal TMA with a total of 608 samples that was built from 8 samples (from 8 different donors) from each of 76 different normal tissue types, as well as cancer TMAs. The cancer TMAs included a total of 18,001 primary tumors from 139 different tumor types and subtypes. The TMAs contained one 0.6 mm spot per tumor and were manufactured as described earlier [30,31]. A histopathological and molecular database was attached to the TMAs containing data of cancers of the breast (n = 1208), kidney (n = 1534), liver (n = 231), bladder (n = 1073), thyroid (n = 518), colon (n = 2351), pancreas (n = 598), stomach (n = 327), endometrium (n = 182), and ovary (n = 369). Clinical follow-up data were available from 877 patients with invasive breast carcinomas of no special type with a median follow-up time of 43 months. From an earlier study, data on PD-L1 expression were available for 11,938 cancers and on the density of CD8-positive lymphocytes for 6,897 cancers [32]. Details on the normal tissues and cancers represented in the TMAs are given in Section 3. All samples were obtained from the routine archives of the Institute of Pathology, University Hospital of Hamburg, Germany; the Institute of Pathology, Clinical Center Osnabrueck, Germany; and the Department of Pathology, Academic Hospital Fuerth, Germany. The use of archived remnants of diagnostic tissues for TMA manufacturing, their analysis for research purposes, and the use of patient data complied with local laws (HmbKHG, §12) and analysis had been approved by the local ethics committee (Ethics commission Hamburg, WF-049/09). All work has been carried out in compliance with the *Declaration of Helsinki*.

### 2.2. Immunohistochemistry (IHC)

All experiments were carried out on the same day and with the same batch of reagents. For best immunostaining results, all TMA blocks were freshly cut one day in advance of immunostaining. Slides were deparaffinized with xylol, rehydrated through a graded alcohol series, and exposed to heat-induced antigen retrieval for 5 minutes in an autoclave at 121 °C in pH 7.8 Tris-EDTA-Citrate (TEC) buffer. Endogenous peroxidase activity was blocked with Dako REAL Peroxidase-Blocking Solution (Agilent Technologies, Santa Clara, CA, USA; #S2023) for 10 minutes. Primary antibody specific for STING (mouse monoclonal, MSVA-515M, MS Validated Antibodies, Hamburg, Germany; #5473-515M) was applied at 37 °C for 60 minutes at a dilution of 1:150. For the purpose of antibody validation, the normal tissue TMA was also analyzed by the rabbit recombinant monoclonal STING antibody clone D2P2F (Cell Signaling Technologies^®^, Danvers, MA, USA; #13647) at a dilution of 1:600 and an otherwise identical protocol. Bound antibody was visualized using the Dako REAL EnVision Detection System Peroxidase/DAB+ Rabbit/Mouse kit (Agilent Technologies, Santa Clara, CA, USA; #K5007), according to the manufacturer’s directions. The sections were counterstained with hemalaun. Scoring of tumor samples was performed as described before [33]. In brief, the percentage of positive neoplastic cells was estimated, and the staining intensity was semi-quantitatively recorded (0, 1+, 2+, or 3+). For statistical analyses, the staining results were categorized into four groups (see Appendix A). Tumors without any staining were considered negative. Tumors with 1+ staining intensity in ≤70% of tumor cells and 2+ intensity in ≤30% of tumor cells were considered weakly positive. Tumors with 1+ staining intensity in >70% of tumor cells, 2+ intensity in 31–70%, or 3+ intensity in ≤30% of tumor cells were considered moderately positive. Tumors with 2+ intensity in >70% or 3+ intensity in >30% of tumor cells were considered strongly positive. We used this scoring system in many earlier TMA studies and found it suitable for the identification of numerous known and novel prognostic molecular features in various tumor types [33].

### 2.3. Statistics

The JMP17^®^ software package (SAS^®^, Cary, NC, USA) was used for statistical analyses including the chi^2^-test to search for associations between STING immunostaining and tumor phenotype and PD-L1 immunostaining in tumor and immune cells, analysis of variance (ANOVA) to search for associations between STING immunostaining and the density of CD8-positive lymphocytes, and the Log-rank test along with Kaplan–Meier plots for survival analysis.

## 3. Results

### 3.1. Technical Issues

In the tumor TMAs, a total of 15,345 (85.2%) of 18,001 tumor samples were interpretable in our analysis. In the normal tissue TMA, at least four samples were evaluable of each normal tissue type. Non-interpretable samples demonstrated an absence of specific cell types, absence of unequivocal tumor cells, or a complete lack of individual tissue spots.

### 3.2. STING in Normal Tissues

STING immunostaining of cells was always cytoplasmic. Particularly strong STING staining was observed in endothelial cells of vessels of all sizes; macrophages/dendritic cells; subsets of lymphocytes and of bone marrow cells, respiratory, fallopian tube, and endocervical epithelium; as well as in basal cells of the prostate. The few cell types that were always negative in our screening of up to eight samples per tissue type included heart and skeletal muscle cells; syncytiotrophoblast and cytotrophoblast cells; amnion and chorion cells of the placenta; epithelial cells of the parathyroid, thyroid, and adrenal glands; acinar cells of the pancreas and of the prostate; hepatocytes; principal cells of the caput epididymis; and Sertoli cells and germ cells of the testis. Representative images of these tissues are shown in Figure 1. In many other cell types, the STING staining pattern varied between samples. For example, STING staining ranged from negative to strong in the gastrointestinal epithelium, gallbladder, collecting ducts of the kidney, urothelium, epithelial cells of the cauda epididymis and the seminal vesicle, and epithelial and stromal cells of the endometrium. STING staining varied from negative to moderate intensity in the salivary glands, Brunner glands, breast epithelium, and in subsets of cells in the adeno- and neurohypophysis. In squamous epithelium, STING staining was usually limited to the basal cell layer, although STING staining was more intense in tonsil crypts. All these normal tissue findings were obtained by using the mouse monoclonal antibody MSVA-515M and the rabbit recombinant monoclonal antibody D2P2F and were, therefore, considered to be specific. Representative images of tissue staining are given in Appendix A.

### 3.3. STING Expression in Cancer

In tumor samples, STING immunostaining was seen in macrophages, lymphocytic cells, endothelial cells, other stroma cells, and (often) also in tumor cells. STING positivity of tumor cells was detectable in 8908 (58.1%) of the 15,345 analyzable tumors, including 4169 (27.2%) with weak, 2005 (13.1%) with moderate, and 2734 (17.8%) with strong immunostaining. Overall, 130 of 139 tumor categories showed detectable STING staining, while 96 tumor categories included at least one case with strong positivity (Table 1). Representative images of STING-positive tumors are shown in Figure 2. Particularly high rates of STING positivity occurred in SQCCs of different sites of origin (up to 96%), malignant mesothelioma (88.5–95.7%), ductal adenocarcinoma of the pancreas (94.9%), pulmonary adenocarcinoma (90.3%), cervical adenocarcinoma (90.0%), serous high-grade ovarian cancer (86.0%), anaplastic thyroid carcinoma (82.9%) colorectal adenocarcinoma (75.2%), adenocarcinoma of the gallbladder (68.8%), and in breast carcinoma (up to 66%). The prevalence of STING positivity was intermediate for urothelial neoplasms (51.3–79.6%), adenocarcinoma of the esophagus (57.4%), and several important sarcoma categories such as liposarcoma (50.9%) and osteosarcoma (48.3%). Particularly low rates of STING positivity were observed for prostatic adenocarcinomas (up to 11%), different subtypes of renal cell carcinomas (RCC; 18.8–24.5%), neuroendocrine tumors (NETs) of various sites (17.0–28.6%), neuroendocrine carcinomas (NECs; 12.5–38.5%), and Merkel cell carcinoma of the skin (5%) and small cell neuroendocrine carcinomas of the bladder (31.6%) and the prostate (17.6%). A graphical representation of a ranking order of STING-positive and strongly positive cancers is given in Appendix A. 

### 3.4. STING Expression, Tumor Phenotype, and Prognosis

The relationship between STING expression in tumor cells and clinically important histopathological and molecular tumor features in carcinomas from different sites is shown in Table 2. In clear cell renal cell carcinoma (ccRCC), high STING expression was linked to a poor histologic grade (*p* < 0.005), high pT category (*p* < 0.0001), and high UICC stage (*p* = 0.0060). In colorectal adenocarcinoma, high STING expression was linked to right-side tumor location (*p* = 0.0008), microsatellite instability (*p* = 0.0016), and RAS mutations (*p* < 0.0001). High STING expression was also related to nodal metastases in hepatocellular carcinoma (HCC; *p* = 0.0435) and in papillary carcinoma of the thyroid (*p* = 0.0074). In invasive breast cancer of no special type, high STING expression was linked to estrogen receptor (ER) and progesterone receptor (PR) expression (*p* < 0.0001 each), non-triple-negative status (*p* = 0.0028), and poor overall survival (*p* = 0.0196; Figure 3). However, in non-invasive urothelial carcinoma of the urinary bladder, high STING expression was associated with a low grade (*p* = 0.0002). A combined analysis of 480 SQCC from nine different sites showed a significant link between STING positivity and HPV infection (*p* = 0.0212; Appendix A). This statistical relationship was also retained in a subgroup of 50 pharyngeal SQCC (*p* = 0.0390). STING expression was unrelated to tumor phenotype in gastric adenocarcinoma, high-grade serous and endometrioid ovarian cancer, endometrioid endometrial carcinoma, pancreatic adenocarcinoma, and papillary RCC (Appendix A).

### 3.5. STING Expression, PD-L1 Status, and Tumor Microenvironment

Data on PD-L1 status were available from 10,579 and on the density of CD8 positive lymphocytes from 5880 tumors for which data on STING expression on tumor cells were collected in our project. Across all tumor entities, there was a significant relationship between high STING expression in tumor cells and PD-L1 positivity of tumor cells and tumor-associated inflammatory cells (*p* < 0.0001 each), while STING expression was unrelated to the density of CD8 positive cells (*p* = 0.4253, Table 3). Examples of PD-L1 and CD8 positive and negative immunostainings are shown in Appendix A.

## 4. Discussion

The main goal of our study was to provide a comprehensive overview of the prevalence of STING expression in tumor cells across a broad range of different tumor entities. Our data from 15,345 tumors from 139 tumor entities demonstrate substantial heterogeneity of tumoral STING expression between cancer types and individual patients. A total of >90% of the analyzed tumor entities had at least one STING positive case, >65% tumor entities had at least one case with strong STING positivity, >50% of all tumors were STING-positive, and 18% of all tumors were strongly STING-positive, demonstrating that STING expression is a common feature of cancer cells. It is still conspicuous that some cancer entities such as SQCCs of various sites and colorectal, pancreatic, gallbladder, pulmonary, or cervical adenocarcinomas did often show high STING expression levels while other entities such as prostatic adenocarcinomas, renal cell carcinomas, or neuroendocrine neoplasm were less frequently STING-positive. Considering that STING expression can be stimulated by the accumulation of cytosolic DAMPs, which are often found in cancer cells with a high degree of genomic instability (summarized in [34]), it could be hypothesized that cancer entities with a lesser degree of genomic instability such as prostatic adenocarcinomas [35], renal cell carcinomas [36], or neuroendocrine neoplasm [37] may be less prone to STING expression than cancers with high levels of genomic alterations such as SQCCs [38] or pulmonary [39] and pancreatic adenocarcinomas [40].

The high variability of STING expression in different samples from identical normal tissue types found in this study demonstrates that the level of STING expression fluctuates in non-neoplastic cell types. Although the reason for this variability remains elusive, it is conspicuous that many of the tissues and cell types that never showed any STING expression in our study, such as the placenta, testicular tubules, or endocrine organs, are particularly well protected from infection by viruses or bacteria as well as other causes of inflammation. The obvious inconsistency of STING expression levels in normal cell types makes it difficult to distinguish whether cancers do up- or downregulate STING, as compared to normal tissue. Most functional studies on STING in cancer have suggested that upregulation may be the prevalent mechanism (summarized in [6]). However, 42% of our tumor samples were completely STING-negative, with numerous STING-positive cells observed in the adjacent tumor stroma. This suggests that suppression of STING expression may occur in a subset of cancers.

The significant associations between high STING expression and poor overall survival in breast cancer, poor histological grade, and advanced stage in ccRCC as well as nodal metastases in HCC and papillary thyroid carcinoma suggests that STING upregulation rather than downregulation in tumor cells tends to be a feature of aggressive cancers. This observation fits well with reports of a possible tumor-promoting role of STING in cancer cells, although the molecular basis is not yet understood [41]. For example, Bakhoum et al. [42] reported that genetically instable tumors can activate STING-dependent noncanonical NF-κB signaling that facilitates metastasis. Alternatively, it is also possible that the upregulation of STING in cancer cells may represent a consequence of progressive dedifferentiation. The fact that loss of STING expression correlated with grade progression in non-invasive urothelial carcinoma shows that—depending on tumor type—the role of STING in determining the aggressiveness of tumor cells may vary. The absence of associations between STING immunostaining and parameters of cancer aggressiveness in gastric adenocarcinomas, high-grade serous and endometrioid ovarian cancer, endometrioid endometrial carcinoma, pancreatic adenocarcinoma, and papillary RCC further demonstrates that the level of STING expression in tumor cells is not likely to represent a pivotal general feature of cancer aggressiveness. Earlier studies analyzing the prognostic impact of STING expression in tumor cells have found inconsistent results. A relationship between high STING expression and unfavorable tumor phenotype or poor prognosis has been found in SQCC of the tongue [16] and—in agreement with our data—in ccRCC [23] and adenocarcinoma of the colon [20], while associations between low STING expression and poor prognosis were described in non-small cell lung carcinoma [27], gastric cancer [15], small cell lung carcinoma [28], and head and neck carcinomas [25]. Other authors could not find significant associations with patient prognosis or tumor phenotype in colorectal cancer [14], SQCC of the head and neck [43], or early gastric neoplastic lesions [44]. 

STING expression levels were also related to several critical molecular features of tumors. The particularly high rate of STING positivity in HPV-positive SQCC is consistent with the role of STING as a sensor for the presence of cytosolic PAMPs. In addition, data from The Cancer Genome Atlas (TCGA) also showed higher STING expression on RNA and protein levels in HPV-positive versus HPV-negative SQCC of the head and neck [45]. However, studies have shown that this correlation is maybe HPV-type specific as HPV18 E7 and HPV16 E7 lead to inhibition of the cGAS/STING pathway and reduced IFN production [46,47]. The significant link of high STING expression to MSI and RAS mutations in this study is also consistent with previous studies examining STING expression in colorectal and lung carcinomas [27,48,49], although some authors have found more RAS mutations in colorectal cancers with STING expression loss [48]. The relationship between STING positivity and ER/PR loss in breast cancer is consistent with recent studies showing elevated expression and activation of STING by DNA damage in ER/PR-negative breast cancer cell lines [50,51]. Another study found that higher STING expression in ER-positive breast cancers was associated with favorable prognosis [52]. 

The continuous increase in PD-L1 expression in both tumor cells and tumor-associated inflammatory cells with increasing levels of STING expression in tumor cells represents a strong confirmation of studies showing a functional relationship between STING activation and PD-L1 upregulation in cancer cells. For example, PD-L1 upregulation by STING-dependent activation of TBK1 was observed in HCC [9] and in HPV-positive cervical cancer cells [53]. Vasiyani et al. showed a STING-mediated, NF-kB-induced upregulation of PD-L1 in triple-negative breast cancer cells [50], and Grabosch et al. found PD-L1 upregulation by STING activation in an ovarian cancer mouse model [54]. Based on the assumption that STING-expressing tumor cells can produce type I IFN, a more intense population of these tumors by tumor-infiltrating lymphocytes was to be expected. The fact that we could not find a significant relationship between the level of STING expression and the density of CD8-positive, tumor-infiltrating immune cells, although data were available from more than 5500 tumors, strongly argues against functionally relevant type I IFN production in these tumors. This is consistent with data from studies exposing STING-expressing tumor cells to the STING agonist cGAMP or dsDNA and finding only low or even absent IFN-β production in mice with chronic lymphatic leukemia and colon cancer cell lines [55,56]. However, others have found that IFN-ß production is independent of cGAS-STING signaling in mismatch-repair-deficient colon cancer cells [43].

Given the high number of tumors analyzed in our study, we focused on thoroughly validating our STING IHC assay. The International Working Group for Antibody Validation (IWGAV) recommends that acceptable antibody validation for IHC on formalin-fixed tissues should involve either a comparison of results from two different independent antibodies or a comparison with expression data from another independent method [57]. Due to the high variability and cell-type specificity of STING expression, comparison with a method based on disaggregated tissue is not ideal for validating the STING antibody. Crucial evidence for the validity of our assay comes from the confirmation of all STING-positive cell types, including staining variabilities within individual tissue samples observed by MSVA-515M and by the independent second antibody D2P2F. It is noteworthy that using a broad selection of normal tissues (n = 76) for antibody validation increases the likelihood of detecting cross-reactivities, since virtually all proteins found in normal human adult cells are included in the validation process.

## 5. Conclusions

Our data provide an overview on the prevalence of STING expression in cancer cells across 139 different tumor types. The highest STING positivity rates occurred in squamous cell carcinomas, malignant mesothelioma, and adenocarcinomas of various origins. Comparison with the tumor phenotype demonstrates that high STING expression rather than STING deficiency tends to be linked to a more aggressive cancer phenotype. High STING expression was linked to adverse phenotypes in breast cancer, clear cell renal cell carcinoma, colorectal adenocarcinoma, hepatocellular carcinoma, and papillary carcinoma of the thyroid. Moreover, our data demonstrate that STING expression in tumor cells is tightly linked to PD-L1 expression, but it may not exert a measurable impact on the quantity of tumor-infiltrating inflammatory cells.

## Figures and Tables

**Figure 1 cancers-16-02425-f001:**
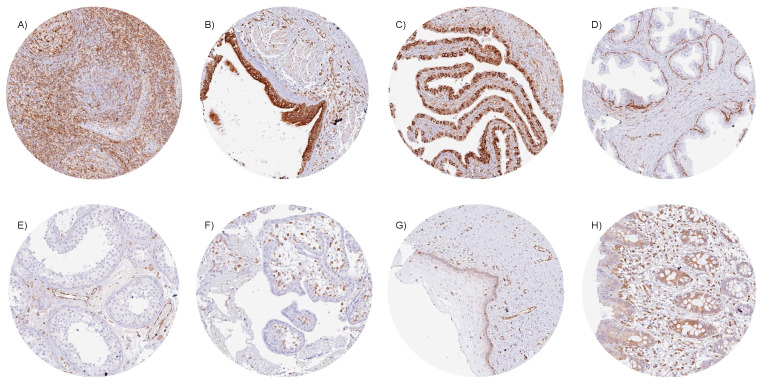
STING immunostaining of normal tissues. (**A**) Strong cytoplasmic staining of many different cell types in the lymph node, (**B**) respiratory epithelial cells of the bronchus, (**C**) a large subset of epithelial cells of the fallopian tube, (**D**) basal cells of the prostate, (**E**) endothelial cells of the testis, and (**F**) stroma cells in the first trimester placenta. (**G**) STING staining is less intense and limited to the basal cell layer in the squamous epithelium of the ectocervix. (**H**) Focal staining in epithelial cells of the rectum.

**Figure 2 cancers-16-02425-f002:**
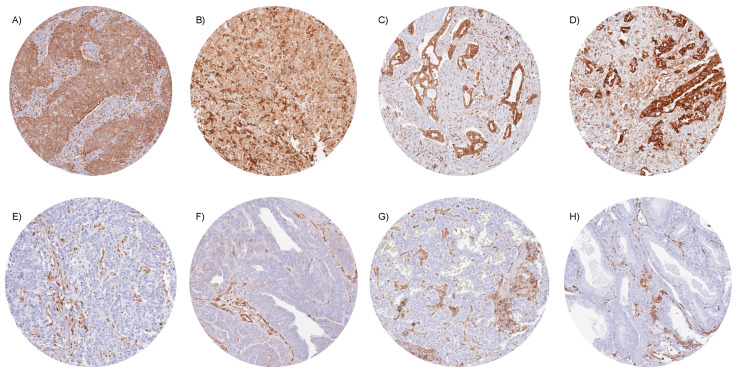
STING immunostaining in cancer. In all tumor samples, cytoplasmic STING staining was seen in stroma cells, including endothelial and inflammatory cells. STING expression was more variable in tumor cells. Strong, cytoplasmic STING positivity for all tumor cells in (**A**) squamous cell carcinoma of the penis, (**B**) epithelioid malignant mesothelioma, (**C**) ductal adenocarcinoma of the pancreas, and (**D**) adenocarcinoma of the lung. STING staining was absent in tumor cells of (**E**) invasive urothelial carcinoma of the urinary bladder, (**F**) serous high-grade carcinoma of the ovary, (**G**) neuroendocrine tumor of the pancreas, and (**H**) adenocarcinoma of the prostate.

**Figure 3 cancers-16-02425-f003:**
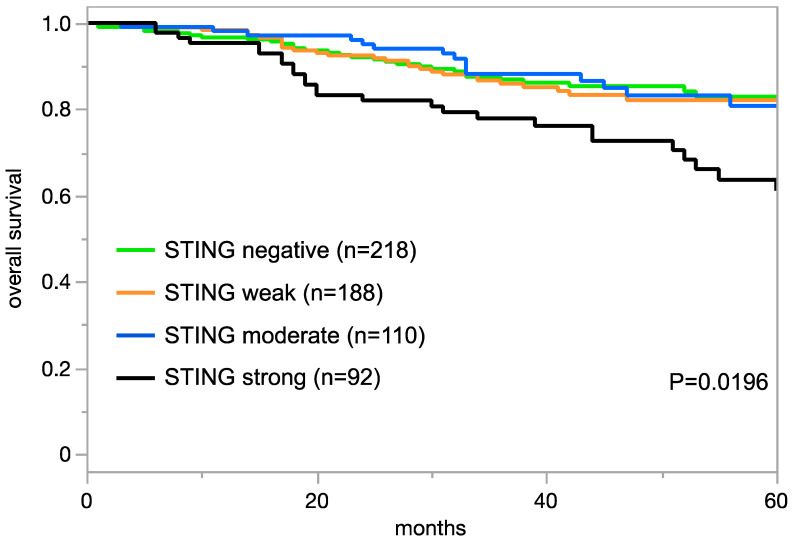
STING immunostaining and prognosis in invasive breast cancer of no special type.

**Table 1 cancers-16-02425-t001:** STING immunostaining in human tumors.

			STING Immunostaining				
Tumor Category	Tumor Entity	on TMA (n)	Analyzable (n)	Negative (%)	Weak (%)	Moderate (%)	Strong (%)
Tumors of the skin	Pilomatricoma	35	23	43.5	21.7	17.4	17.4
	Basal cell carcinoma of the skin	89	58	8.6	53.4	20.7	17.2
	Benign nevus	29	25	4.0	44.0	44.0	8.0
	Squamous cell carcinoma of the skin	145	129	28.7	45.7	16.3	9.3
	Malignant melanoma	65	60	28.3	18.3	28.3	25.0
	Malignant mel. lymph node metastasis	86	69	27.5	30.4	21.7	20.3
	Merkel cell carcinoma	48	40	95.0	2.5	0.0	2.5
Tumors of the head and neck	Squamous cell carcinoma of the larynx	109	94	24.5	39.4	19.1	17.0
	Squamous cell carcinoma of the pharynx	60	52	3.8	40.4	23.1	32.7
	Oral squamous cell carcinoma	130	112	19.6	35.7	23.2	21.4
	Pleomorphic adenoma (parotid gland)	50	42	4.8	9.5	21.4	64.3
	Warthin tumor of the parotid gland	49	46	34.8	58.7	6.5	0.0
	Basal cell adenoma of the salivary gland	15	14	14.3	57.1	14.3	14.3
Tumors of the lung, pleura, and thymus	Adenocarcinoma of the lung	196	165	9.7	18.2	20.0	52.1
	Squamous cell carcinoma of the lung	80	58	25.9	27.6	15.5	31.0
	Small cell carcinoma of the lung	16	12	75.0	0.0	25.0	0.0
	Mesothelioma, epithelioid	40	26	11.5	38.5	34.6	15.4
	Mesothelioma, biphasic	77	46	4.3	19.6	30.4	45.7
	Thymoma	29	27	7.4	37.0	22.2	33.3
	Lung, neuroendocrine tumor (NET)	29	28	71.4	10.7	7.1	10.7
Tumors of the female genital tract	Squamous cell carcinoma of the vagina	78	63	12.7	42.9	19.0	25.4
	Squamous cell carcinoma of the vulva	157	135	23.0	47.4	16.3	13.3
	Squamous cell carcinoma of the cervix	136	123	9.8	34.1	22.0	34.1
	Adenocarcinoma of the cervix	23	20	10.0	25.0	20.0	45.0
	Endometrioid endometrial carcinoma	338	278	22.3	31.7	19.8	26.3
	Endometrial serous carcinoma	86	66	27.3	34.8	12.1	25.8
	Carcinosarcoma of the uterus	57	48	22.9	45.8	18.8	12.5
	Endometrial carcinoma, high grade, G3	13	11	54.5	27.3	9.1	9.1
	Endometrial clear cell carcinoma	9	7	57.1	42.9	0.0	0.0
	Endometrioid carcinoma of the ovary	130	115	20.9	25.2	26.1	27.8
	Serous carcinoma of the ovary	580	520	14.0	29.4	16.5	40.0
	Mucinous carcinoma of the ovary	101	81	45.7	27.2	13.6	13.6
	Clear cell carcinoma of the ovary	51	47	59.6	21.3	6.4	12.8
	Carcinosarcoma of the ovary	47	45	11.1	68.9	6.7	13.3
	Granulosa cell tumor of the ovary	44	41	46.3	39.0	14.6	0.0
	Leydig cell tumor of the ovary	4	4	50.0	25.0	25.0	0.0
	Sertoli cell tumor of the ovary	1	1	0.0	100.0	0.0	0.0
	Sertoli–Leydig cell tumor of the ovary	3	3	66.7	33.3	0.0	0.0
	Steroid cell tumor of the ovary	3	3	66.7	33.3	0.0	0.0
	Brenner tumor	41	37	43.2	45.9	2.7	8.1
Tumors of the breast	Invasive breast ca. of no special type	1764	1564	34.0	32.4	17.5	16.0
	Lobular carcinoma of the breast	363	274	35.8	33.6	19.3	11.3
	Medullary carcinoma of the breast	34	28	35.7	32.1	10.7	21.4
	Tubular carcinoma of the breast	29	19	47.4	42.1	10.5	0.0
	Mucinous carcinoma of the breast	65	52	57.7	26.9	5.8	9.6
	Phyllodes tumor of the breast	50	33	66.7	21.2	12.1	0.0
Tumors of the digestive system	Adenomatous polyp, low-grade dysplasia	50	37	2.7	24.3	24.3	48.6
	Adenomatous polyp, high-grade dyspl.	50	40	5.0	15.0	30.0	50.0
	Adenocarcinoma of the colon	2483	2245	24.8	34.4	15.5	25.3
	Gastric adenocarcinoma, diffuse type	215	151	43.0	35.1	13.2	8.6
	Gastric adenocarcinoma, intestinal type	215	188	39.9	32.4	10.6	17.0
	Gastric adenocarcinoma, mixed type	62	49	30.6	42.9	16.3	10.2
	Adenocarcinoma of the esophagus	83	68	42.6	36.8	13.2	7.4
	Squamous cell carcinoma of the esophagus	76	58	27.6	34.5	15.5	22.4
	Squamous cell carcinoma of the anal canal	91	68	11.8	36.8	27.9	23.5
	Cholangiocarcinoma	58	58	60.3	17.2	6.9	15.5
	Gallbladder adenocarcinoma	51	48	31.3	33.3	12.5	22.9
	Carcinoma of the extrahepatic bile duct	42	31	22.6	19.4	19.4	38.7
	Hepatocellular carcinoma	312	276	84.4	7.2	3.6	4.7
	Ductal adenocarcinoma of the pancreas	659	594	5.1	17.0	26.4	51.5
	Pancreatic/ampullary adenocarcinoma	98	94	13.8	23.4	24.5	38.3
	Acinar cell carcinoma of the pancreas	18	18	77.8	16.7	5.6	0.0
	Gastrointestinal stromal tumor (GIST)	62	62	32.3	16.1	22.6	29.0
	Appendix, neuroendocrine tumor (NET)	25	13	76.9	15.4	0.0	7.7
	Colorectal, neuroendocrine tumor (NET)	12	9	77.8	0.0	22.2	0.0
	Ileum, neuroendocrine tumor (NET)	53	49	100.0	0.0	0.0	0.0
	Pancreas, neuroendocrine tumor (NET)	101	88	83.0	5.7	6.8	4.5
	Colorectal, neuroendocrine carcinoma (NEC)	14	13	76.9	15.4	7.7	0.0
	Ileum, neuroendocrine carcinoma (NEC)	8	8	87.5	0.0	12.5	0.0
	Gallbladder, neuroendocrine carcinoma (NEC)	4	4	75.0	25.0	0.0	0.0
	Pancreas, neuroendocrine carcinoma (NEC)	14	13	61.5	30.8	0.0	7.7
Tumors of the urinary system	Non-invasive papillary urothelial carcinoma, pTa G2 low grade	177	113	20.4	69.0	8.0	2.7
	Non-invasive papillary urothelial carcinoma, pTa G2 high grade	141	100	37.0	50.0	6.0	7.0
	Non-invasive papillary urothelial carcinoma, pTa G3	219	152	48.7	40.1	8.6	2.6
	Urothelial carcinoma, pT2-4 G3	735	517	42.6	30.4	12.6	14.5
	Squamous cell carcinoma of the bladder	22	19	21.1	63.2	5.3	10.5
	Small cell neuroendocrine carcinoma of the bladder	23	19	68.4	21.1	5.3	5.3
	Sarcomatoid urothelial carcinoma	25	17	29.4	47.1	23.5	0.0
	Urothelial carcinoma of the kidney pelvis	62	52	44.2	46.2	3.8	5.8
	Clear cell renal cell carcinoma	1287	1065	75.5	15.7	3.6	5.3
	Papillary renal cell carcinoma	368	320	81.3	13.4	4.1	1.3
	Clear cell tubulopapillary renal cell carcinoma	26	21	76.2	9.5	14.3	0.0
	Chromophobe renal cell carcinoma	170	149	79.2	18.1	2.7	0.0
	Oncocytoma of the kidney	257	219	59.4	34.2	5.0	1.4
Tumors of the male genital organs	Adenocarcinoma of the prostate, Gleason 3+3	83	73	97.3	2.7	0.0	0.0
	Adenocarcinoma of the prostate, Gleason 4+4	80	62	88.7	9.7	1.6	0.0
	Adenocarcinoma of the prostate, Gleason 5+5	85	77	92.2	6.5	0.0	1.3
	Adenocarcinoma of the prostate (recurrence)	258	224	91.5	6.3	1.8	0.4
	Small cell neuroendocrine carcinoma of the prostate	19	17	82.4	11.8	5.9	0.0
	Seminoma	682	659	96.2	3.6	0.2	0.0
	Embryonal carcinoma of the testis	54	47	100.0	0.0	0.0	0.0
	Leydig cell tumor of the testis	31	27	51.9	37.0	11.1	0.0
	Sertoli cell tumor of the testis	2	2	100.0	0.0	0.0	0.0
	Sex cord stromal tumor of the testis	1	1	0.0	100.0	0.0	0.0
	Spermatocytic tumor of the testis	1	1	100.0	0.0	0.0	0.0
	Yolk sac tumor	53	43	95.3	4.7	0.0	0.0
	Teratoma	53	44	43.2	45.5	9.1	2.3
	Squamous cell carcinoma of the penis	92	73	8.2	43.8	28.8	19.2
Tumors of endocrine organs	Adenoma of the thyroid gland	113	111	88.3	9.9	1.8	0.0
	Papillary thyroid carcinoma	391	349	22.1	17.8	15.8	44.4
	Follicular thyroid carcinoma	154	142	78.9	13.4	1.4	6.3
	Medullary thyroid carcinoma	111	102	57.8	27.5	14.7	0.0
	Parathyroid gland adenoma	43	29	96.6	3.4	0.0	0.0
	Anaplastic thyroid carcinoma	45	41	17.1	26.8	24.4	31.7
	Adrenal cortical adenoma	48	45	100.0	0.0	0.0	0.0
	Adrenal cortical carcinoma	27	20	100.0	0.0	0.0	0.0
	Pheochromocytoma	51	51	100.0	0.0	0.0	0.0
Tumors of hematopoietic and lymphoid tissues	Hodgkin’s lymphoma	103	95	1.1	14.7	42.1	42.1
	Small lymphocytic lymphoma, B-cell type	50	44	13.6	84.1	0.0	2.3
	Diffuse large B cell lymphoma (DLBCL)	113	100	28.0	57.0	8.0	7.0
	Follicular lymphoma	88	74	6.8	77.0	13.5	2.7
	T-cell non-Hodgkin’s lymphoma	25	21	4.8	38.1	4.8	52.4
	Mantle cell lymphoma	18	13	0.0	92.3	0.0	7.7
	Marginal zone lymphoma	16	14	14.3	78.6	7.1	0.0
	Diffuse large B-cell lymphoma (DLBCL) in the testis	16	16	25.0	68.8	6.3	0.0
	Burkitt lymphoma	5	1	100.0	0.0	0.0	0.0
Tumors of soft tissue and bone	Tenosynovial giant cell tumor	45	31	3.2	3.2	12.9	80.6
	Granular cell tumor	53	31	100.0	0.0	0.0	0.0
	Leiomyoma	50	47	83.0	14.9	2.1	0.0
	Leiomyosarcoma	94	86	38.4	30.2	15.1	16.3
	Liposarcoma	145	114	49.1	21.9	14.0	14.9
	Malignant peripheral nerve sheath tumor (MPNST)	15	13	23.1	30.8	23.1	23.1
	Myofibrosarcoma	26	24	29.2	25.0	16.7	29.2
	Angiosarcoma	74	55	7.3	25.5	20.0	47.3
	Angiomyolipoma	91	75	5.3	12.0	22.7	60.0
	Dermatofibrosarcoma protuberans	21	15	6.7	26.7	6.7	60.0
	Ganglioneuroma	14	14	7.1	71.4	14.3	7.1
	Kaposi sarcoma	8	5	0.0	20.0	20.0	60.0
	Neurofibroma	117	112	38.4	49.1	12.5	0.0
	Sarcoma, not otherwise specified (NOS)	74	64	31.3	29.7	12.5	26.6
	Paraganglioma	41	39	97.4	2.6	0.0	0.0
	Ewing sarcoma	23	11	36.4	9.1	36.4	18.2
	Rhabdomyosarcoma	7	6	50.0	16.7	0.0	33.3
	Schwannoma	122	113	23.0	50.4	15.0	11.5
	Synovial sarcoma	12	9	88.9	11.1	0.0	0.0
	Osteosarcoma	44	29	51.7	27.6	10.3	10.3
	Chondrosarcoma	40	24	70.8	12.5	4.2	12.5
	Rhabdoid tumor	5	5	60.0	0.0	20.0	20.0
	Solitary fibrous tumor	17	16	12.5	37.5	37.5	12.5

**Table 2 cancers-16-02425-t002:** STING immunostaining and tumor phenotype.

Tumor Entity	Pathological and Molecular Parameters		STING Immunostaining				
		n	Negative (%)	Weak (%)	Moderate (%)	Strong (%)	*p*
Invasive breast carcinoma of no special type	pT1	543	35.5	30.6	20.3	13.6	0.2402
	pT2	419	33.9	34.6	17.4	14.1	
	pT3-4	84	32.1	28.6	15.5	23.8	
	G1	161	30.4	34.8	21.1	13.7	0.3571
	G2	541	35.1	29.4	19.4	16.1	
	G3	384	36.5	33.9	15.6	14.1	
	pN0	337	33.1	29.0	20.9	17.0	0.1367
	pN+	435	34.4	35.0	17.8	12.7	
	pM0	168	38.7	25.0	20.8	15.5	0.4778
	pM1	97	34.0	34.0	17.5	14.4	
	HER2 negative	831	34.5	32.1	18.5	14.8	0.0746
	HER2 positive	114	37.7	40.4	13.2	8.8	
	ER negative	196	46.4	32.7	9.2	11.7	<0.0001
	ER positive	709	31.0	34.3	20.0	14.7	
	PR negative	384	41.9	32.8	12.5	12.8	<0.0001
	PR positive	561	29.9	32.8	21.2	16.0	
	Non-triple negative	748	31.7	34.8	19.4	14.2	0.0028
	Triple negative	130	47.7	30.0	10.8	11.5	
Clear cell renal cell carcinoma	ISUP 1	232	81.9	14.2	1.3	2.6	0.0002
	ISUP 2	352	79.0	12.2	4.3	4.5	
	ISUP 3	231	66.2	19.5	4.3	10.0	
	ISUP 4	66	63.6	21.2	7.6	7.6	
	Fuhrman 1	56	85.7	8.9	3.6	1.8	0.0002
	Fuhrman 2	603	79.8	13.3	3.3	3.6	
	Fuhrman 3	259	68.7	19.3	3.5	8.5	
	Fuhrman 4	81	58.0	24.7	7.4	9.9	
	Thoenes 1	306	82.4	13.1	2.0	2.6	0.0017
	Thoenes 2	424	72.6	16.3	3.8	7.3	
	Thoenes 3	87	63.2	20.7	5.7	10.3	
	UICC 1	270	79.3	14.1	2.6	4.1	0.0060
	UICC 2	31	67.7	25.8	0.0	6.5	
	UICC 3	82	65.9	15.9	2.4	15.9	
	UICC 4	58	65.5	13.8	8.6	12.1	
	pT1	593	81.6	12.3	3.2	2.9	<0.0001
	pT2	114	71.9	19.3	3.5	5.3	
	pT3-4	297	65.7	19.5	4.7	10.1	
	pN0	147	70.7	15.6	4.8	8.8	0.5488
	pN+	23	60.9	13.0	8.7	17.4	
	pM0	86	70.9	18.6	4.7	5.8	0.7933
	pM+	79	69.6	15.2	6.3	8.9	
Hepatocellular carcinoma	pT1	67	83.6	6.0	4.5	6.0	0.3890
	pT2	74	74.3	14.9	2.7	8.1	
	pT3-4	56	82.1	5.4	7.1	5.4	
	G1	36	88.9	5.6	0.0	5.6	0.2932
	G2	113	76.1	9.7	5.3	8.8	
	G3	46	84.8	8.7	4.3	2.2	
	pN0	66	75.8	7.6	7.6	9.1	0.0435
	pN+	35	48.6	22.9	11.4	17.1	
Papillary carcinoma of the thyroid	pT1	135	19.3	19.3	23.0	38.5	0.0961
	pT2	73	31.5	15.1	12.3	41.1	
	pT3-4	94	20.2	18.1	11.7	50.0	
	pN0	82	23.2	19.5	22.0	35.4	0.0074
	pN+	119	9.2	16.8	16.8	57.1	
Urothelial bladder carcinoma	pTa G2 low	113	20.4	69.0	8.0	2.7	0.0002
	pTa G2 high	100	37.0	50.0	6.0	7.0	
	pTa G3	131	47.3	41.2	9.2	2.3	
	pT2	103	41.7	26.2	15.5	16.5	0.4343
	pT3	189	37.6	31.7	13.8	16.9	
	pT4	91	46.2	34.1	9.9	9.9	
	G2	18	33.3	33.3	27.8	5.6	0.2402 *
	G3	373	41.3	30.8	12.3	15.6	
	pN0	224	39.7	30.4	12.5	17.4	0.3742 *
	pN+	146	44.5	31.5	13.0	11.0	
Adenocarcinoma of the colon	pT1	82	18.3	28.0	23.2	30.5	0.1202
	pT2	427	24.4	33.7	18.3	23.7	
	pT3	1200	25.7	34.5	13.6	26.3	
	pT4	425	25.4	35.5	16.5	22.6	
	pN0	1125	25.7	32.5	15.1	26.7	0.1591
	pN+	999	24.2	36.3	16.0	23.4	
	V0	1535	25.6	32.9	15.7	25.8	0.2577
	V1	557	22.6	37.3	15.3	24.8	
	L0	691	25.6	31.5	15.2	27.6	0.1674
	L1	1411	24.5	35.7	15.7	24.0	
	Right side	445	21.1	30.6	18.0	30.3	0.0008
	Left side	1196	27.1	35.6	14.6	22.7	
	MMR proficient	1124	25.0	35.7	16.1	23.2	0.0016
	MMR deficient	83	16.9	22.9	19.3	41.0	
	RAS wildtype	445	33.0	33.3	15.7	18.0	<0.0001
	RAS mutation	331	17.2	31.4	18.7	32.6	
	BRAF wildtype	125	26.4	32.0	16.8	24.8	0.5262
	BRAF V600E mutation	19	15.8	36.8	10.5	36.8	

* only in pT2-4 urothelial bladder carcinomas. Abbreviations: pT: pathological tumor stage, G: grade, pN: pathological lymph node status, pM: pathological status of distant metastasis, V: venous invasion, L: lymphatic invasion, PR: progesterone receptor, MMR: mismatch repair, ER: estrogen receptor, ISUP: International Society of Urological Pathology, UICC: Union for International Cancer Control.

**Table 3 cancers-16-02425-t003:** STING immunostaining versus PD-L1 immunostaining and density of CD8-positive lymphocytes.

	PD-L1-Positive (% of Tumors)				CD8^+^ Density (Cells/mm^2^)	
STING Immunostaining	n	Tumor Cells	n	Immune Cells	n	Mean ± SE
negative	4142	8.0	4136	27.2	1893	287.6 ± 11.6
weak	2767	14.2	2763	33.8	1615	270.9 ± 12.5
moderate	1318	19.5	1316	32.7	862	266.1 ± 17.2
strong	1799	23.2	1796	37	1222	298.4 ± 14.1
*p*		<0.0001		<0.0001		0.4253

Abbreviations: SE: standard error.

## Data Availability

All data generated or analyzed during this study are included in this published article.

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
