# Peer review of "Stimulator of Interferon Genes Protein (STING) Expression in Cancer Cells: A Tissue Microarray Study Evaluating More than 18,000 Tumors from 139 Different Tumor Entities"

_cancers, 2024, doi:10.3390/cancers16132425_

Round 1

Reviewer 1 Report

Comments and Suggestions for Authors

Anne Menz et al., reported in this manuscript “Stimulator of interferon genes protein (STING) expression in cancer cells: A tissue microarray study evaluating more than 18,000 tumors from 139 different tumor entities” a unique catalog of STING expression in tumor cells as well as its clinical relevance in more than 130 different tumor entities. The authors identified the expression of STING paralleled PD-L1 positivity of tumor and inflammatory cells but was unrelated to the density of CD8+ lymphocytes. Overall, the story sounds interesting. However, the article needs to be revised before it is published:

1. Result 3.1 is an explanation, not a result, and should be placed in the Materials and Methods.

2. Please provide figures of the experimental results of PD-L1 positive tissue and CD8+positive tissue, not just statistical results.

3. Figure 3: The color differentiation of the lines should be more pronounced.

4. As is well known, STING is involved in immune responses, and STING agonists are currently evaluated as anticancer drugs in the field of cancer immunotherapy. However, Why is high expression of STING associated with adverse tumor phenotype and poor overall survival?

5. Recent evidence indicates that the combination of STING agonists with STAT3 inhibitors can enhance tumor immunogenicity and optimize the immunotherapeutic effects, and thus effectively enhanced anti-tumor immunity (PubMed ID: 32972405). Activation of the IL6/STAT3 signaling pathway promote cancer metastasis and progression (PubMed ID: 30918019). Please add this aspect to the discussion.

Comments on the Quality of English Language

The manuscript's English is generally clear, yet certain sentences could benefit from increased conciseness to enhance readability. For instance, in the abstract, the phrase "STING is an important activator of immune responses in inflammatory cells" can be streamlined to "STING activates immune responses in inflammatory cells."

The language style throughout the manuscript is formal and professional. However, some sections are excessively complex. It is advised to preserve the formal and professional tone while improving readability by simplifying sentence structures.

Author Response

Anne Menz et al., reported in this manuscript “Stimulator of interferon genes protein (STING) expression in cancer cells: A tissue microarray study evaluating more than 18,000 tumors from 139 different tumor entities” a unique catalog of STING expression in tumor cells as well as its clinical relevance in more than 130 different tumor entities. The authors identified the expression of STING paralleled PD-L1 positivity of tumor and inflammatory cells but was unrelated to the density of CD8+ lymphocytes. Overall, the story sounds interesting. However, the article needs to be revised before it is published:

  1. Result 3.1 is an explanation, not a result, and should be placed in the Materials and Methods.

Reply: We rephrased 3.1 to make it clearer that it represents results.

  1. Please provide figures of the experimental results of PD-L1 positive tissue and CD8+positive tissue, not just statistical results.

Reply: We added examples of PD-L1 and CD8 positive and negative tumors in Supplementary Figure 3.

  1. Figure 3: The color differentiation of the lines should be more pronounced.

Reply: We have changed the colors of the lines.

  1. As is well known, STING is involved in immune responses, and STING agonists are currently evaluated as anticancer drugs in the field of cancer immunotherapy. However, Why is high expression of STING associated with adverse tumor phenotype and poor overall survival?

Reply: We discussed possible tumor promoting functions of STING upregulation on page 15, lines 291-296)

  1. Recent evidence indicates that the combination of STING agonists with STAT3 inhibitors can enhance tumor immunogenicity and optimize the immunotherapeutic effects, and thus effectively enhanced anti-tumor immunity (PubMed ID: 32972405). Activation of the IL6/STAT3 signaling pathway promote cancer metastasis and progression (PubMed ID: 30918019). Please add this aspect to the discussion.

Reply: As suggested by the reviewer, we add this aspect on page 2, lines 55-56.

Comments on the Quality of English Language

The manuscript's English is generally clear, yet certain sentences could benefit from increased conciseness to enhance readability. For instance, in the abstract, the phrase "STING is an important activator of immune responses in inflammatory cells" can be streamlined to "STING activates immune responses in inflammatory cells."

The language style throughout the manuscript is formal and professional. However, some sections are excessively complex. It is advised to preserve the formal and professional tone while improving readability by simplifying sentence structures.

Reply: We have simplified the language where appropriate.

Submission Date

28 May 2024

Date of this review

09 Jun 2024 07:25:42

Reviewer 2 Report

Comments and Suggestions for Authors

Very nice paper in which plenty of performed work is well visible. I don’t have any particular methodological concerns. Moreover references are properly chosen. 
There are some minor flaws which I would recommend to take into consideration prior publication which this paper undoubtedly deserves.
1. what was the rationale of exactly such quantification? Any reference? System of evaluation?
2. It would be much more informative if evaluation values would be presented in the table.
3. In general term “tumor tissue” does not exist and shouldn’t be used…

4. Biggest concern and technical issue is number of numerical data/results… there is for sure much to much numbers… there should be some clarification or exclusion of many of a data, probably in replacing the, into supplemental data or even separate paper… at least authors should focus on most important findings and place them in more visible manner, or even graphs etc… 

5. I would for sure expect more comprehensive conclusion/ expanded hypothesis/ any take home message, because at present it is missing part.

Above mentioned points are in regard to the whole image of the revised paper small expected amendments.
In my opinion they are very easy to introduce making possible publication relatively fast track process. 

Author Response

Comments and Suggestions for Authors

Very nice paper in which plenty of performed work is well visible. I don’t have any particular methodological concerns. Moreover references are properly chosen. 
There are some minor flaws which I would recommend to take into consideration prior publication which this paper undoubtedly deserves. 

  1. what was the rationale of exactly such quantification? Any reference? System of evaluation? 2. It would be much more informative if evaluation values would be presented in the table.

Reply: We have now better explained our scoring system on page 3, lines 137-139 and added suppl. table 1 to present the evaluation values in a table. This scoring system proved to be efficient in more than 200 TMA studies by our group and others and enabled finding all known associations between molecular markers and tumor phenotype and patient prognosis.

  1. In general term “tumor tissue” does not exist and shouldn’t be used…

Reply: We have replaced “tumor tissue” by “tumor” where appropriate.

  1. Biggest concern and technical issue is number of numerical data/results… there is for sure much to much numbers… there should be some clarification or exclusion of many of a data, probably in replacing the, into supplemental data or even separate paper… at least authors should focus on most important findings and place them in more visible manner, or even graphs etc… 

Reply: We very much agree with the reviewer that there is a lot of numerical data. In reply to reviewer’s concern, we have emphazised this in the discussion and better explained the rational of the study on page 15, lines 256-259.

  1. I would for sure expect more comprehensive conclusion/ expanded hypothesis/ any take home message, because at present it is missing part.

Reply: We have expanded the conclusion part (page 16, last para) to be more comprehensive.

Above mentioned points are in regard to the whole image of the revised paper small expected amendments. 
In my opinion they are very easy to introduce making possible publication relatively fast track process. 

Submission Date

28 May 2024

Date of this review

16 Jun 2024 15:23:17
